# SARS-CoV-2 reliably detected in frozen saliva samples stored up to one year

Jennifer K. Frediani [1,2], Kaleb B. McLendon [2,3], Adrianna Westbrook [2,4], Scott E. Gillespie [4], Anna Wood [4], Tyler J. Baugh [2,3], William O'Sick [2,3], John D. Roback [2,3], Wilbur A. Lam [2,5], Joshua M. Levy [2,6]*

1 Nell Hodgson Woodruff School of Nursing, Emory University, Atlanta, GA, United States of America, 2 The Atlanta Center for Microsystems-Engineered Point-of-Care Technologies, Atlanta, GA, United States of America, 3 Emory/Children's Laboratory for Innovative Assay Development, Atlanta, GA, United States of America, 4 Department of Pediatrics, Emory University School of Medicine, Atlanta, GA, United States of America, 5 Aflac Cancer and Blood Disorders Center at Children's Healthcare of Atlanta, Department of Pediatrics, Emory University School of Medicine, Atlanta, GA, United States of America, 6 Department of Otolaryngology-Head and Neck Surgery, Emory University School of Medicine, Atlanta, GA, United States of America

* joshua.levy2@emory.edu

**Data Availability Statement:** All relevant data are within the paper and its Supporting Information files.

**Funding:** This work was supported by the National Institute of Biomedical Imaging and Bioengineering

## Abstract

Viability of saliva samples stored for longer than 28 days has not been reported in the literature. The COVID-19 pandemic has spawned new research evaluating various sample types, thus large biobanks have been started. Residual saliva samples from university student surveillance testing were retested on SalivaDirect and compared with original RT-PCR (cycle threshold values) and quantitative antigen values for each month in storage. We conclude that saliva samples stored at -80°C are still viable in detecting SARS-CoV-2 after 12 months of storage, establishing the validity of these samples for future testing.

## Introduction

Saliva samples are utilized in various diagnostic assays, including COVID-19 [1–4]. During the COVID-19 pandemic, large biobanks were started to provide clinical samples for several different studies, including pivotal validation assays needed to obtain regulatory clearance for new diagnostic tests. Clinical samples (i.e., nasopharyngeal, anterior nares and mid turbinate swabs, and saliva) collected from patients with known viral status are used as positive and negative controls in the evaluation of test performance, such as sensitivity and specificity. These samples are collected within a standardized timeframe of symptom onset and are often stored and shipped prior to use in the evaluation of a new diagnostic test under controlled conditions.

However, several factors, such as the longitudinal stability of patient samples, remain uncertain. In this study we focus on human saliva as a frequently utilized sample type for the evaluation of emerging COVID-19 diagnostic tests.

The results of sample testing are closely evaluated by regulators as part of the evaluation process for determining if a new test is safe for public use. In the U.S., these data are reviewed by the FDA prior to awarding an Emergency Use Authorization or other approval for public

RADx program, [Grant Number: 54 EB027690 02S1] WL and the National Center for Advancing Translational Sciences [Grant Number: UL1 TR002378](PI not an author). Funders had no role in study design, data collection and analysis, decision to publish, or preparation of the manuscript.

**Competing interests:** The authors have declared that no competing interests exist.

utilization. However, it is currently unknown if sample quality, defined as the reliability of a previously positive sample to remain positive after storage, is affected by storage time. The goal of this study is therefore to evaluate the longitudinal stability of saliva samples collected from patients with SARS-CoV-2 in hopes of determining if storage time influences the quality of the collected sample.

## Methods

### Sample collection and analysis

Deidentified residual samples were originally collected from Emory University faculty, staff, and students (asymptomatic surveillance testing) beginning in November of 2020. This study was in compliance with all research guidelines and due to the nature of residual samples did not require IRB submission. Specimens were collected using the SalivaDirect unsupervised collection kit (Yale School of Public Health, New Haven, CT, USA) [5, 6], including a short straw Salimetrics Saliva Collection Aid,(Salimetrics, LLC, Carlsbad, CA, USA, catalog #5016.02) and a sterile 2 mL plastic tube containing 3 ceramic beads. For ease of pipetting, samples were homogenized at 4.5 m/s for 5 seconds using the Omni International Bead Ruptor Elite (Omni International, owned by Perkin Elmer, Kennesaw, GA, USA). Specimens were screened for nucleocapsid antigen via Quanterix HD-X using the SARS-CoV-2 N Protein Advantage assay (Quanterix Corporation, Billerica, MA, USA). Saliva residuals were stored at 4˚C during Quanterix antigen screening. Samples resulting positive for antigen (cutoff >/ = 3.00pg/mL) were then reflexed to confirmatory PCR. Saliva samples were tested according to the SalivaDirect dual-plexed RT-qPCR protocol. This testing was performed on the QuantStudio platform (ThermoFisher Scientific, Waltham, MA, USA) [7], using reagents and materials qualified for the SalivaDirect procedure. PCR+ specimens were reported, stored at -80˚C, then archived using OpenSpecimen software (Krishnagi LLC, St. Louis, MO, USA). When retesting sample residuals for this study, the same protocols were repeated. All samples went through a single freeze-thaw cycle with completion of re-testing in November 2021.

### Sample selection

We selected 10 random samples from each month, December 2020 to October 2021. If a month had less than 10 samples available, then all samples were used. If a month had more than 10 samples available, we selected an additional 4–5 random samples to be used if any of the original chosen 10 could not be re-tested. There was one case where the original sample selected nor any alternate was available due to lack of volume in the residual sample. Some months did not have any positive samples due to most students living off campus at the time for either the entire month or part of the month (i.e., May and June 2021). In some cases, the residual amount was only enough for PCR retesting and antigen testing was not completed. All samples were re-tested on the same day (11/18/21) and therefore we are evaluating the viability from one month old up to 12 months old.

### Statistical analysis

Participants who had an "undetermined" N1 cycle threshold (Ct) test result due to having a Ct above a detectable threshold were considered to have an N1 Ct of 40, the detectable threshold, for the main analysis. Likewise, participants with an antigen concentration below the detectable limit were assigned an antigen concentration of 0.098 pg/mL, directly below the limit of detection, for the main analysis. Participants with other non-numerical results were excluded from the main analysis. Participants who were given an imputed Ct value of 40 and/or an

imputed antigen concentration of 0.098 pg/mL for the main analysis were removed for subsequent sensitivity analyses.

The test and re-test N1 Ct values and antigen concentrations were compared over the entire study period and by month. To gauge the differences between the test and re-test values, the average absolute difference (re-test minus test) and percent change (re-test minus test divided by test) were calculated. A coefficient of variation for test and re-test values were calculated for each month to increase comparability between samples. Pearson correlation was also measured to determine the correlation between test values. Intraclass correlations (ICC) were calculated to determine the reliability between results. An ICC close to one represents high reliability while an ICC close to zero represents low reliability and indicates freezing degrades the samples and distorts true N1 Ct values and/or antigen concentrations. All analyses we re-performed in sensitivity analyses excluding imputed observations. Data management and correlation measures were performed in SAS 9.4 (Cary, NC). Intraclass correlation was calculated with SPSS version 28 (IBM, Armonk, NY). Graphs were created in R v4.1.3 (Vienna, Austria).

## Results

A total of 87 de-identified Emory-affiliated participants were included in study analysis.

Overall, the mean re-test N1 Ct decreased from the test value but maintained high correlation and reliability in the main analysis (Mean Ct value from original test 95% CI 27.7 [26.6, 28.8]; mean re-test Ct value 95% CI 26.9 [25.6, 28.2]; -2.9% change; Pearson correlation 95% CI 0.92 [0.87, 0.94]; Intraclass Correlation (ICC) 95% CI 0.90 [0.84, 0.94]) (Table 1). In a sensitivity analysis including only participants with complete N1 Ct data, the correlation and reliability decreased to moderate values (Mean Ct value from original test 26.7; mean re-test Ct value 25.6; -4.1% change; Pearson correlation 95% CI 0.89 [0.84, 0.93]; ICC 95% CI 0.87 [0.72, 0.93]) (S1 Table). As shown in Fig 1, most thawed specimens maintained a similar Ct to that seen in the original sample. Eight out of nine months demonstrated high correlation and reliability (0.75–0.90), with six of those months displaying excellent reliability (>0.90), indicating that saliva samples can maintain stability even with several months of storage at -80 (Fig 2). January, the second oldest samples, displayed low correlation and reliability. This same trend was observed in the sensitivity analysis since no January observations were removed in the sensitivity analysis (Mean Ct value from original test 28.2; mean re-test Ct value 26.3; -6.7% change; Pearson correlation 95% CI 0.42 [-0.43, 0.86]; ICC 95% CI 0.37 [-0.25, 0.82]). Through visual assessment, ICC appears to remain relatively stable across the year, except for January, indicating that ICC does not decrease with increasing time and that swabs can maintain the same level of stability with increasing time.

Antigen concentrations displayed variable correlations and reliability with no clear trend over time (Mean antigen concentration from original test 95% CI 333.3 [-42.7, 709.3]; mean re-test antigen concentration 95% CI 178.9 [88.0, 269.8]; -43.3% change; Pearson correlation 95% CI 0.80 [0.68, 0.88]; Intraclass correlation (ICC) 95% CI 0.37 [0.11, 0.58]) (Table 2). It should be noted that the mean concentrations, although all considered positive, changed widely from month to month which could explain the lack of a clear trend. Sensitivity analyses excluding observations that were below the limit of detection showed similar results (Mean antigen concentration from original test 95% CI 352.6 [-45.5, 750.8]; mean re-test antigen concentration 95% CI 189.5 [93.9, 285.1]; -46.3% change; Pearson correlation 95% CI 0.81 [0.68, 0.88]; Intraclass correlation (ICC) 95% CI 0.37 [0.10, 0.58]) (S2 Table). While N1 Ct values appear to remain stable over long periods of being frozen, antigen concentrations seem to lack that same reliability. However, test and re-test antigen concentrations maintain a positive correlation throughout time and at some points display moderate agreement.

**Table 1. Test and re-test Ct value differences by month.**

| Month | N | Mean Test N1 CT (95% CI) | Mean Re-test N1 CT (95% CI) | Absolute Difference (95% CI) | Test Coefficient of Variation | Re-test Coefficient of Variation | Percent Change | Pearson Correlation (95% CI) | Intraclass Correlation (95% CI) |
|---|---|---|---|---|---|---|---|---|---|
| Overall | 87 | 27.7 (26.6, 28.8) | 26.9 (25.6, 28.2) | -0.8 (-1.3, -0.3) | 19.2 | 22.5 | -2.9% | 0.92 (0.87, 0.94) | 0.90 (0.84, 0.94) |
| December '20 | 10 | 30.8 (26.7, 34.8) | 29.5 (24.9, 34.2) | -1.3 (-3.4, 0.9) | 18.5 | 22.1 | -4.2% | 0.89 (0.56, 0.97) | 0.87 (0.59, 0.97) |
| January '21 | 8 | 28.2 (25.4, 30.9) | 26.3 (24.1, 28.5) | -1.8 (-4.5, 0.9) | 11.6 | 10.1 | -6.7% | 0.42 (-0.43, 0.86) | 0.37 (-0.25, 0.82) |
| February '21 | 10 | 27.9 (23.7, 32.2) | 27.9 (22.6, 33.3) | 0.0 (-2.6, 2.5) | 21.3 | 26.8 | 0.0% | 0.88 (0.54, 0.97) | 0.87 (0.57, 0.97) |
| March '21 | 10 | 29.0 (25.9, 32.1) | 28.4 (24.7, 32.1) | -0.6 (-1.8, 0.6) | 14.8 | 18.0 | -2.1% | 0.95 (0.80, 0.99) | 0.94 (0.79, 0.98) |
| April '21 | 10 | 27.8 (24.7, 30.9) | 27.3 (23.3, 31.2) | -0.5 (-1.9, 0.9) | 15.7 | 20.2 | -1.8% | 0.95 (0.76, 0.99) | 0.92 (0.74, 0.98) |
| July '21 | 9 | 26.3 (22.7, 30.0) | 25.5 (21.1, 29.8) | -0.8 (-2.5, 0.7) | 18.1 | 22.2 | -3.0% | 0.93 (0.67, 0.98) | 0.91 (0.68, 0.98) |
| August '21 | 10 | 25.7 (21.2, 30.2) | 25.1 (20.5, 29.6) | -0.6 (-2.3, 1.1) | 24.5 | 25.5 | -2.3% | 0.93 (0.69, 0.98) | 0.93 (0.76, 0.98) |
| September '21 | 10 | 25.0 (20.9, 29.0) | 24.5 (19.9, 29.1) | -0.5 (-1.9, 1.1) | 22.7 | 26.3 | -2.0% | 0.95 (0.77, 0.99) | 0.94 (0.80, 0.99) |
| October '21 | 10 | 28.5 (24.2, 32.9) | 27.3 (22.0, 32.6) | -1.2 (-2.5, 0.1) | 21.2 | 27.1 | -4.2% | 0.98 (0.93, 0.996) | 0.95 (0.78, 0.99) |

## Discussion

This study is the first to examine the validity of saliva samples after long term storage for detection of SARS-CoV-2. Positivity remained stable up to 12 months at -80˚C. While nasopharyngeal samples are the gold standard, saliva has become a more prominent sample type for its ease of collection and storage possibilities. While there is not an approved lateral flow assay for saliva currently in the US, this is an active area of research, and several are available globally.

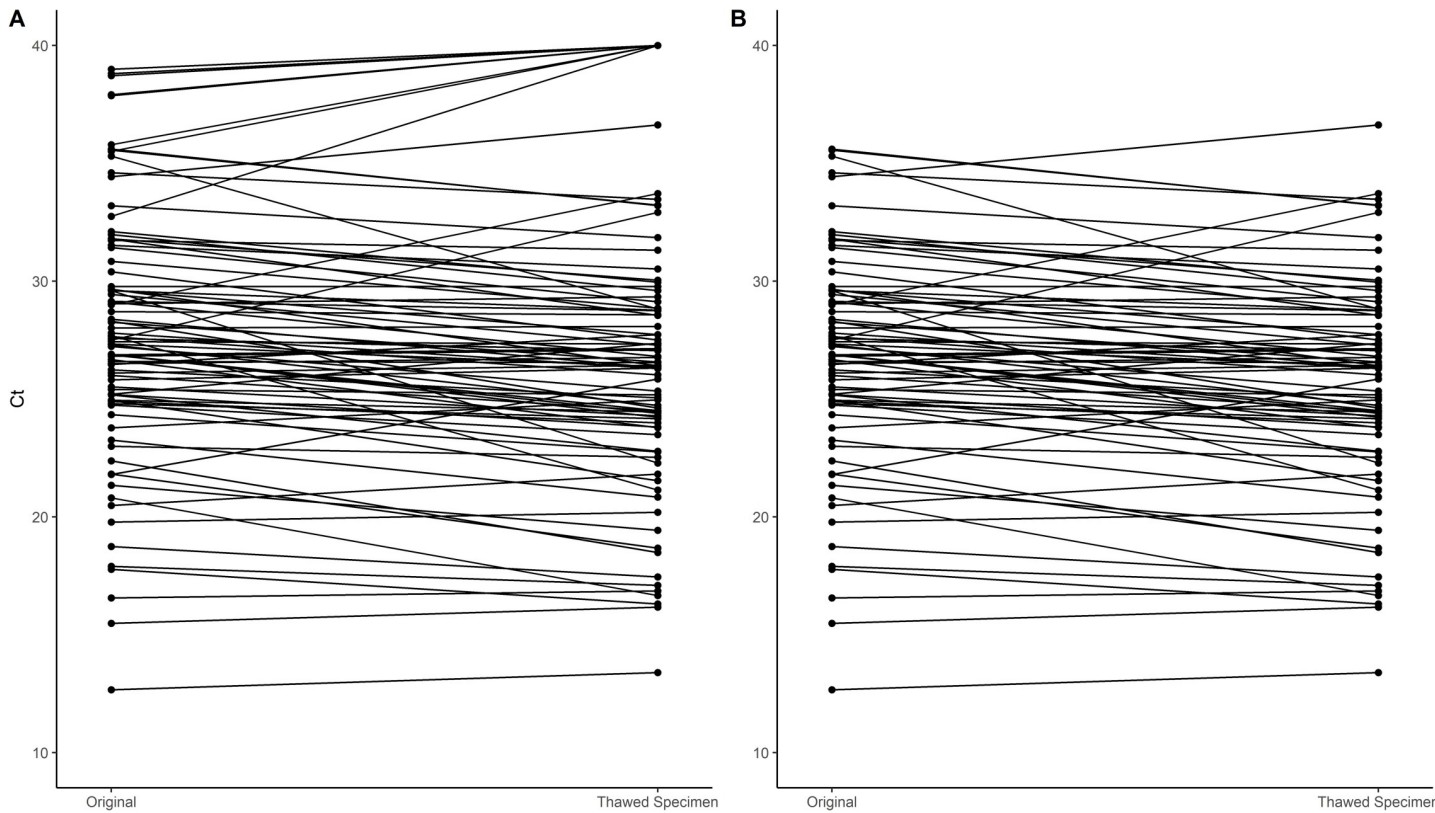

**Fig 1. Plot of original and thawed specimen Ct values with lines connecting observations from the same individual.** Panel A includes all participants involved in the main analysis while Panel B shows only those individuals included in the sensitivity analysis by excluding those with a Ct above 40. Most samples maintained a similar re-thaw Ct to that seen in the original sample.

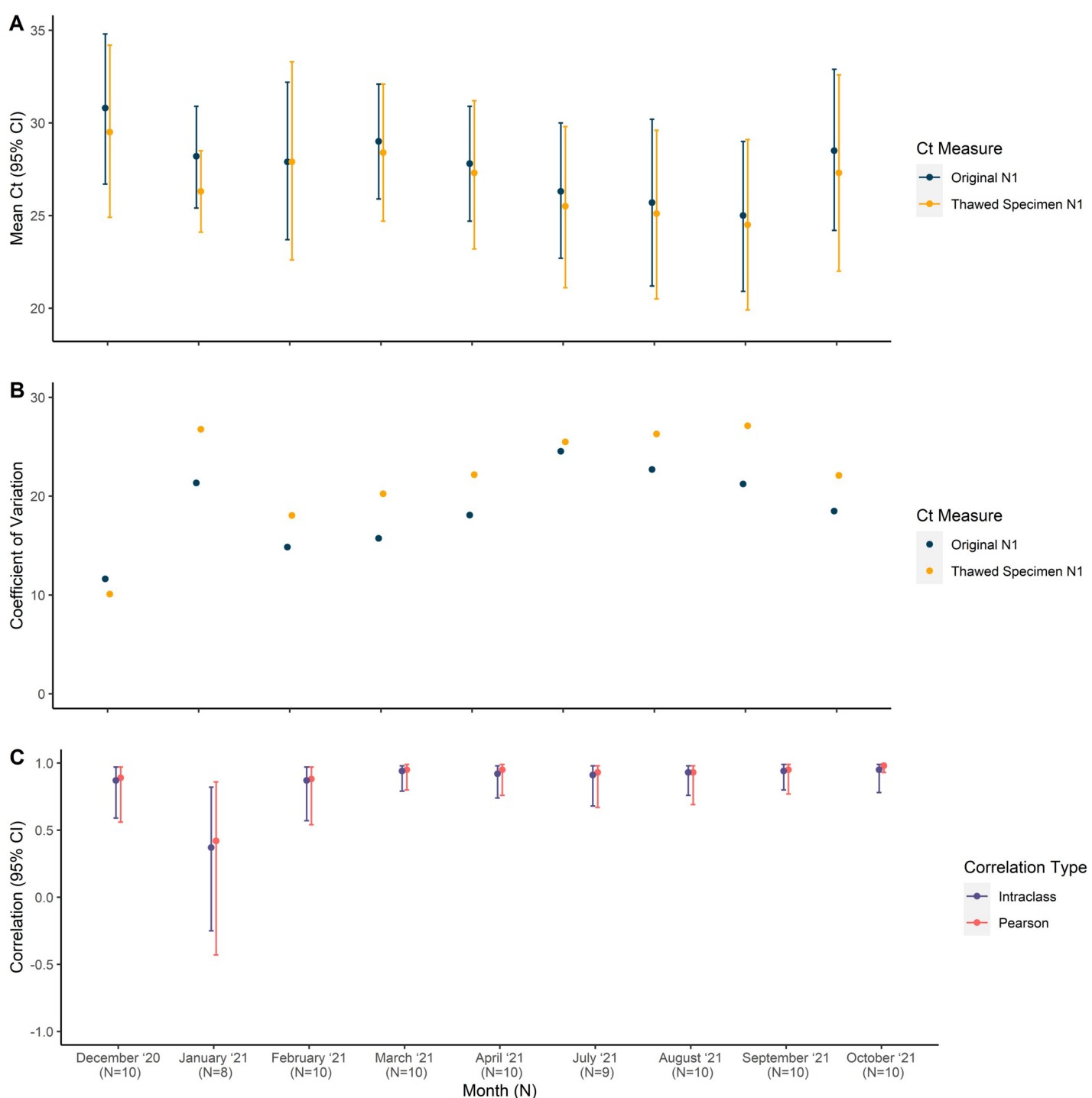

**Fig 2. Test and re-test Ct values vary over time but remain highly correlated, except for January.** A) The mean Ct values were highest in December 2020 and then decreased over the next 8 months before rising again in September 2021. B) The coefficient of variation across the study period stayed consistently below 30 and were similar within each month. C) The test and re-test Ct values are strongly correlated (Pearson correlation > 0.70) and demonstrate good reliability (Intraclass correlation > 0.75) across the entire study period except for January, indicating that frozen saliva swabs can remain reliable over long periods of time.

Most studies of saliva stability for diagnostic test validation have examined the stability of cold chain operations from time of collection to time of analysis, therefore using one week to

**Table 2. Test and re-test antigen concentration differences by month.**

| Month | N | Mean Test Antigen Concentration (95% CI) | Mean Re-test Antigen Concentration (95% CI) | Absolute Difference (95% CI) | Test Coefficient of Variation | Re-test Coefficient of Variation | Percent Change | Pearson Correlation (95% CI) | Intraclass Correlation (95% CI) |
|---|---|---|---|---|---|---|---|---|---|
| Overall | 54 | 333.3 (-42.7, 709.3) | 178.9 (88.0, 269.8) | -154.4 (-462.0, 153.2) | 413.3 | 186.2 | -43.3% | 0.80 (0.68, 0.88) | 0.37 (0.11, 0.58) |
| December '20 | 8 | 27.4 (-4.9, 59.7) | 59.1 (-15.9, 134.2) | 31.7 (-12.7, 76.2) | 141.2 | 151.8 | 115.7% | 0.97 (0.81, 0.99) | 0.66 (0.07, 0.92) |
| January '21 | 7 | 1726.8 (-1648.7, 5102.4) | 555.3 (-28.0, 1138.6) | -1171.5 (-4022.6, 1679.6) | 146.6 | 94.7 | -67.8% | 0.92 (0.47, 0.99) | 0.31 (-0.47, 0.83) |
| February '21 | 9 | 244.2 (-30.9, 519.3) | 151.3 (41.2, 261.4) | -92.9 (-286.3, 100.4) | 135.7 | 200.1 | -38.0% | 0.83 (0.33, 0.96) | 0.57 (-0.05, 0.88) |
| March '21 | 8 | 32.3 (-4.3, 68.8) | 45.0 (-30.3, 120.3) | 12.7 (-66.4, 91.9) | 135.7 | 200.1 | 39.3% | 0.13 (-0.64, 0.76) | 0.12 (-0.73, 0.74) |
| April '21 | 9 | 53.6 (10.5, 96.8) | 47.2 (15.7, 78.7) | -6.4 (-39.4, 26.5) | 104.7 | 86.8 | -11.9% | 0.65 (-0.06, 0.91) | 0.64 (-0.01, 0.91) |
| July '21 | 5 | 102.6 (-44.4, 249.5) | 280.5 (-207.0, 768.0) | 177.9 (-191.4, 547.2) | 115.4 | 140.0 | 173.4% | 0.86 (-0.21, 0.99) | 0.44 (-0.37, 0.92) |
| August '21 | 7 | 318.3 (-365.4, 1001.9) | 236.3 (-157.9, 630.5) | -82.0 (-403.1, 239.1) | 232.3 | 180.4 | -25.8% | 0.96 (0.74, 0.99) | 0.85 (0.37, 0.97) |
| October '21 | 1 | 12.0 (--) | 97.0 (--) | 85 (--) | -- | -- | 708.3% | -- | -- |

NOTE: Because there was 1 observation in October '21, it was not possible to calculate certain values which is denoted by --.

one month time frames. For comparison, Perchetti et.al. studied the stability of nasopharyngeal samples transported in phosphate-buffered saline at various temperatures over 28 days. All samples remained stable at -80˚C, while concentrations varied more with lower temperatures (i.e., room temperature, 4˚C, and -20˚C) [8]. Ahannach et.al. and Alfaro-Núñez et.al. both determined viability of saliva swab samples, dipped in a collected saliva or inoculated from an oropharyngeal sample, respectively [9, 10]. Ahannach et.al. stored samples at -4˚C for 3 weeks, then for 3 days at either room temperature, -20˚C, or -80˚C before DNA extraction for microbiome analysis. Taxonomic composition did not differ between storage temperature [9]. Alfaro-Núñez et.al. tested saliva samples kept at room temperature, 4˚C and -20˚C for SARS-CoV-2 at 25 days from collection. Ct values were more stable within 9 days and out to 25 days at -20˚C. In summary, lower temperatures -20˚C and -80˚C increase stability over time and in our study, -80˚C storage was able to guarantee positivity at 12 months post collection. Ott et al focused on nonsupplemented saliva samples at -80˚C, ~19˚C, and 30˚C for anywhere between 3 and 92 days and found all saliva samples remained stable [11]. Here we studied only -80˚C storage of nonsupplemented saliva, as this is our standard protocol for all biobanked samples.

## Strengths and weaknesses

The strengths of this study include the full year time period. Previous studies only include a few months after collection where we investigate a full year. The limitations of this study include the small sample size at each month. Since these were deidentified residuals we were limited in times where positivity rates were low. In the case of January, we feel this is a statistical issue of small sample size and not a sample handling issue. All documentation for those samples show that the handling of the January samples were correct and the same as the other months. However, since those lab technicians are no longer in the lab, we can only rely on documentation. From a statistical standpoint, January has the smallest number of samples and therefore less precision than other months. Intraclass correlation is a ratio of the variance of interest to the total variance so when there is more variance between the different participant samples due to a lower sample size than between the test and re-test samples, then the ICC is lower. However, one experiment out of nine could be a reflection of noise. Overall results where all months were combined, which allowed us to decrease the statistical noise, showed high reliability and should be given more emphasis than the smallest individual experiment.

## Conclusion

The implication of this research enables COVID-19 researchers and others in the microbiology and virology fields to utilize biobanked saliva samples up to one year for detection of SARS--CoV-2 and possibly other viruses, although further research is needed.

## Supporting information

**S1 File. Deidentified dataset.**
(PDF)

**S1 Table. Test and re-test Ct value differences by month from samples with complete data.**
(DOCX)

**S2 Table. Test and re-test antigen concentration differences by month excluding observations that were below the limit of detection.**
(DOCX)

## Author Contributions

**Conceptualization:** Jennifer K. Frediani, Joshua M. Levy.

**Data curation:** Kaleb B. McLendon, Tyler J. Baugh.

**Formal analysis:** Adrianna Westbrook, Anna Wood.

**Funding acquisition:** Wilbur A. Lam.

**Supervision:** Scott E. Gillespie, William O'Sick, John D. Roback, Wilbur A. Lam, Joshua M. Levy.

**Writing – original draft:** Jennifer K. Frediani.

**Writing – review & editing:** Kaleb B. McLendon, Adrianna Westbrook, Joshua M. Levy.

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
