## [Decision Letter · Decision Letter 0]

20 May 2022

PONE-D-22-12527SARS-CoV-2 reliably detected in frozen saliva samples stored up to one yearPLOS ONE

Dear Dr. Frediani,

Thank you for submitting your manuscript to PLOS ONE. After careful consideration, we feel that it has merit but does not fully meet PLOS ONE’s publication criteria as it currently stands. Therefore, we invite you to submit a revised version of the manuscript that addresses the points raised during the review process.

ACADEMIC EDITOR: As appended below, the reviewers have raised major concerns/critiques (reviewer # 3 is against publication) and suggested further justification/work to consolidate the findings. Do go through the comments and amend the MS accordingly.

We look forward to receiving your revised manuscript.

Kind regards,

A. M. Abd El-Aty

Academic Editor

PLOS ONE

Journal Requirements:

3. Thank you for stating the following in Funding Section of your manuscript:

“This work was supported by the National Institute of Biomedical Imaging and Bioengineering RADx program, [Grant Number: 54 EB027690 02S1] WL,https://www.nibib.nih.gov/, and the National Center for Advancing Translational Sciences [Grant Number: UL1 TR002378](PI not an author),https://ncats.nih.gov/”

” This work was supported by the National Institute of Biomedical Imaging and Bioengineering RADx program, [Grant Number: 54 EB027690 02S1] WL,https://www.nibib.nih.gov/, and the National Center for Advancing Translational Sciences [Grant Number: UL1 TR002378](PI not an author),https://ncats.nih.gov/

Funders had no role in study design, data collection and analysis, decision to publish, or preparation of the manuscript”

Reviewers' comments:

Reviewer's Responses to Questions

**Comments to the Author**

1. Is the manuscript technically sound, and do the data support the conclusions?

Reviewer #1: Partly

Reviewer #2: Partly

Reviewer #3: No

Reviewer #4: Partly

Reviewer #5: Partly

Reviewer #6: Partly

2. Has the statistical analysis been performed appropriately and rigorously? 

Reviewer #1: No

Reviewer #2: I Don't Know

Reviewer #3: N/A

Reviewer #4: No

Reviewer #5: No

Reviewer #6: No

3. Have the authors made all data underlying the findings in their manuscript fully available?

Reviewer #1: Yes

Reviewer #2: Yes

Reviewer #3: Yes

Reviewer #4: Yes

Reviewer #5: Yes

Reviewer #6: Yes

4. Is the manuscript presented in an intelligible fashion and written in standard English?

Reviewer #1: Yes

Reviewer #2: No

Reviewer #3: Yes

Reviewer #4: No

Reviewer #5: Yes

Reviewer #6: No

5. Review Comments to the Author

Reviewer #1: This is an interesting paper which presents some criticisms that should be fixed before its eventual publication

Major criticisms

No data are clearly presented regarding the analytical coefficient of variation of the assay used. Therefore, the eventual decrease after storage cannot be completely evaluated. This issue should be clarified

The figure should be improved as some variations should be better specified according to the previous criticism

Introduction:

the initial sentence seems inappropriate. The assay used is not a POCT and therefore, the real need is to develop accurate laboratory tests, not only POCT

References to be added, if possible:

a) Basso D, Aita A, Padoan A, Cosma C, Navaglia F, Moz S, Contran N, Zambon CF, Maria Cattelan A, Plebani M. Salivary SARS-CoV-2 antigen rapid detection: A prospective cohort study. Clin Chim Acta. 2021 Jun;517:54-59. doi: 10.1016/j.cca.2021.02.014.

b) Basso D, Aita A, Navaglia F, Mason P, Moz S, Pinato A, Melloni B, Iannelli L, Padoan A, Cosma C, Moretto A, Scuttari A, Mapelli D, Rizzuto R, Plebani M. The University of Padua salivary-based SARS-CoV-2 surveillance program minimized viral transmission during the second and third pandemic wave. BMC Med. 2022 Feb 23;20(1):96. doi: 10.1186/s12916-022-02297-1.

Reviewer #2: Abstract, line 65 – I would update that viability has not been reported on, rather than not studied as it is quite possible that it has been studied on in a number of settings but not specifically reported on.

Line 75 – I feel this opening sentence is not well aligned with the overall theme/message of the paper. The investigations in this paper do not involve POC technologies and apply to diagnostic development broader than just POC. I suggest that this introduction be reframed to better introduce the work and message in the paper.

Line 105 – please cite the paper on unsupervised collection devices mentioned here (https://doi.org/10.1186/s12879-022-07285-7)

Line 112 – please cite the paper, EUA or protocols.io on SalivaDirect for reference to this method (or more the#5 citation to the first mention of SalivaDirect on line 112 rather than after the PCR instrument).

Line 164/165 – remove gap between “- 6.7%”

Being a dualplex qPCR, it would be interesting for the authors to also report results for RP over time and how this compares to SARS-CoV-2 detection.

The authors are missing perhaps the earliest work on stability of unsupplemented raw saliva and SARS-CoV-2 detection and are likely more relevant than those currently included in the discussion: doi: 10.3201/eid2704.204199

The citation recommended for line 105 also reports on stability of SARS-CoV-2 RNA when cycled through various temperatures, and demonstrating that cold chain transport is not required.

The figures would be more informative, if in addition to the averages depicted, if the results for each pair could also be depicted. This would allow the reader to more robustly analyses how the pairs performed.

The authors fail to reflect on some of the large changes in Ct values between some of the pairs. Were re-tests double checked? Could anything different have happened during that time (primers, MMX)? Could any samples have not been tested properly the first time? Are any discrepancies more consistent per month perhaps further indicating a slight difference in that first test month?

Line 2 of the supplementary table shows an initial result of 0 – that doesn’t seem to be accurate. It could be helpful to have table either by month or by initial Ct.

Reviewer #3: The authors evaluated the reliability of the detection of SARS-CoV-2 RNA and antigen in frozen saliva samples stored for up to one year. The stability of saliva for the detection of SARS-CoV-2 RNA has been intensively investigated by researchers; however, the methods of preparation, preservation, and examination differed among the laboratories and high-quality investigations are still desired. Unfortunately, the current studies only showed the correlation between the first and second evaluations in a small number of samples, mainly in asymptomatic individuals with a single PCR assay and antigen testing.

Useful information as scientific data for universalization

(1) Sufficient number of samples for evaluation

(2) Stratified data

1) High viral load samples, moderate viral load samples, and low viral load samples

2) Conditions of preservation (4, -20, -80)

3) Data of several molecular assays, including commonly available assays

4) Difference of variants

Overall, the current research is insufficient to use as scientific data. Also, the quality of manuscript is low for an original article.

Minor point

CT - Ct

Line 112: Yale’s SalivaDirect dual-plexed RT-qPCR protocol: please add the reference or link

Line 115: -80C -� -80ºC

Reviewer #4: Enough information is presented (either in the paper or in cited references) to enable other investigators to replicate the work.

The presentation of the results could be improved in several ways, most notably by placing the numerical comparisons in a table, rather than presenting them as text.

The authors have drawn some strange conclusions based on the Ct values for the assays. IN comparing mean Ct values over time, they have not presented confidence intervals for the means, and have made statements about changes which seem likely to me to be “noise.” Furthermore, they seem to suggest that lower Ct values represent lower reliability, which isn’t true at all. They state that “January, the second oldest samples, displayed low correlation and reliability.” While the figure supports this conclusion, the authors provide no explanation for this odd behavior (error in assay performance? Specimen mishandling?).

The Figure could be significantly improved. There is no justification for the lines connecting the point estimates. Each point estimate should be accompanied by confidence intervals.

The authors have described antigen concentration assessment, and presented some verbiage in the results section, without presenting any sort of formal analysis. I suppose the comments are supported by data in the supplementary Excel spreadsheet, but I would be much happier if there were to be a more formal presentation of what the data say.

Reviewer #5: The paper is concise and straightforward, the conclusions are valid. The only part that is missing is alignment with MIQE guidelines for quantitative real-time PCR (DOI: 10.1373/clinchem.2008.112797). Understandably, the authors used commercial kits. However, if the information on validation of the kits is available, it is worth including it into the manuscript.

Another comment, the mean values in Figure 1 are given without error bars and n values for the number of samples. From supplementary data it is hard to understand how many samples were available for retesting for each month.The significance of the difference in the January sample should be reported by p-value.

Reviewer #6: The manuscript submitted by Dr. Frediani and entitled “SARS-CoV-2 reliably detected in frozen saliva samples stored up to one year” evaluates the long-term stability of frozen saliva specimens for the detection of SARS-CoV-2. In this work 87 specimens that were previously positive for SARS-CoV-2 and frozen and -80C were retested. The results of the specimens were then compared to the initial run to look at specimen stability. In general they found stable specimens for PCR with average decreases after the freeze thaw. The results of the study are useful as saliva for respiratory diagnostics is becoming more common due to the pandemic and understanding the matrix’s long-term stability will be valuable for laboratories and industry to validate novel assays when the viral target is low. However, in its current state there are some clarifications needed and re-writing necessary for publication. Here are my suggestions:

Major Comments

Ln132: Why would you create a CT value when no value was obtained. These should just not be run on statistical analysis since giving them a point value. When observing your data in figure 1 and in the averages, the freeze thaw seemed to improve detection based on the lower CT. Would this be more pronounced if the negative specimens were removed. It is also important to discuss these missed samples and the CT from the initial test. Were they near the LoD.

From the data trend, it almost appears that a freeze thaw improves the sensitivity of the assay, which has been a discussion in the field and possible concern for the FDA in evaluating retrospecitive specimens. It would be beneficial if a small subset of new specimens could be frozen and tested after 24 hours of freeze thaw to determine if

Ln124 and throughout: I would suggest reviewing the manuscript for conversational and indirect language. As an example, ln 124 states” the original chosen 10 could not be re-tested for some reason”. I would remove “for some reason” and add a sentence of the numbers that were not tested and reasons. This continues into more of the methods, which make it a bit unclear of how the specimens were tested. For example, n the sample selection when 10 were taken from December and 10 from January are these tested at the same time so one batch is X-months frozen and the January batch is X-1 months old or were they all stored for 12 months prior to testing?

Minor comments

Ln104: What is meant by compliance (i.e. was this approval via the institutions IRB)?

I would consider changing the term re-test to thawed specimens or something similar. When I read re-test I am thinking of a possible repeat for a test that was invalid.

Link figures in text when indicated and presenting data.

As there is only 87 data points I think it would be interesting to see the N1 CT values as a dots where the samples are on X axis CT on Y and 2 points for each sample so we can see the spread of CT values for individual samples and not as an average.

6. PLOS authors have the option to publish the peer review history of their article (what does this mean?). If published, this will include your full peer review and any attached files.

Reviewer #1: No

Reviewer #2: No

Reviewer #3: No

Reviewer #4: No

Reviewer #5: No

Reviewer #6: No

---

## [Author Response · Author response to Decision Letter 0]

5 Jul 2022

Reviewer Comments Author comments

Editor comments Check style requirements (file naming) Completed

 Please provide additional details regarding participant consent. In the ethics statement in the Methods and online submission information, please ensure that you have specified (1) whether consent was informed and (2) what type you obtained (for instance, written or verbal, and if verbal, how it was documented and witnessed). If your study included minors, state whether you obtained consent from parents or guardians. If the need for consent was waived by the ethics committee, please include this information. This has been clarified in the text, due to the nature of these de-identified archived samples no IRB was required. 

 If you are reporting a retrospective study of medical records or archived samples, please ensure that you have discussed whether all data were fully anonymized before you accessed them and/or whether the IRB or ethics committee waived the requirement for informed consent. If patients provided informed written consent to have data from their medical records used in research, please include this information. Added more detail to the current sentence in the Methods section. See above

 Thank you for stating the following in Funding Section of your manuscript:

“This work was supported by the National Institute of Biomedical Imaging and Bioengineering RADx program, [Grant Number: 54 EB027690 02S1] WL,https://www.nibib.nih.gov/, and the National Center for Advancing Translational Sciences [Grant Number: UL1 TR002378](PI not an author),https://ncats.nih.gov/”

” This work was supported by the National Institute of Biomedical Imaging and Bioengineering RADx program, [Grant Number: 54 EB027690 02S1] WL,https://www.nibib.nih.gov/, and the National Center for Advancing Translational Sciences [Grant Number: UL1 TR002378](PI not an author),https://ncats.nih.gov/

Funders had no role in study design, data collection and analysis, decision to publish, or preparation of the manuscript”

Please include your amended statements within your cover letter; we will change the online submission form on your behalf. This has been removed, no changes needed

 We note that you have indicated that data from this study are available upon request. PLOS only allows data to be available upon request if there are legal or ethical restrictions on sharing data publicly. For more information on unacceptable data access restrictions We have included a data file in the supplementary materials.

Reviewer 1 No data are clearly presented regarding the analytical coefficient of variation of the assay used. Therefore, the eventual decrease after storage cannot be completely evaluated. This issue should be clarified

The figure should be improved as some variations should be better specified according to the previous criticism We have substantially updated the figure to address concerns from all reviewers. In this updated figure, you will find that we have added a panel for the coefficient of variation for both the original and re-test values for each month. The coefficient of variation was consistently below 30 for all months and was similar for the original and re-test values within a month. 

 Introduction:

the initial sentence seems inappropriate. The assay used is not a POCT and therefore, the real need is to develop accurate laboratory tests, not only POCT Reworded to reflect the objective of the paper.

 References to be added, if possible:

a) Basso D, Aita A, Padoan A, Cosma C, Navaglia F, Moz S, Contran N, Zambon CF, Maria Cattelan A, Plebani M. Salivary SARS-CoV-2 antigen rapid detection: A prospective cohort study. Clin Chim Acta. 2021 Jun;517:54-59. doi: 10.1016/j.cca.2021.02.014.

b) Basso D, Aita A, Navaglia F, Mason P, Moz S, Pinato A, Melloni B, Iannelli L, Padoan A, Cosma C, Moretto A, Scuttari A, Mapelli D, Rizzuto R, Plebani M. The University of Padua salivary-based SARS-CoV-2 surveillance program minimized viral transmission during the second and third pandemic wave. BMC Med. 2022 Feb 23;20(1):96. doi: 10.1186/s12916-022-02297-1.

 These have been included in the introduction.

Reviewer 2 Abstract, line 65 – I would update that viability has not been reported on, rather than not studied as it is quite possible that it has been studied on in a number of settings but not specifically reported on.

 This has been changed.

 Line 75 – I feel this opening sentence is not well aligned with the overall theme/message of the paper. The investigations in this paper do not involve POC technologies and apply to diagnostic development broader than just POC. I suggest that this introduction be reframed to better introduce the work and message in the paper.

 Reworded to reflect the objective of the paper.

 Line 105 – please cite the paper on unsupervised collection devices mentioned here (https://doi.org/10.1186/s12879-022-07285-7)

Added

 Line 112 – please cite the paper, EUA or protocols.io on SalivaDirect for reference to this method (or more the#5 citation to the first mention of SalivaDirect on line 112 rather than after the PCR instrument). Added

 Line 164/165 – remove gap between “- 6.7%” There isn’t a gap between the – and 6.7%.

 Being a dualplex qPCR, it would be interesting for the authors to also report results for RP over time and how this compares to SARS-CoV-2 detection. For the SalivaDirect protocol, the RP signal is only used to verify that amplification occurred before reporting a negative result. While the authors agree that the RP trends may be interesting, the analysis required to collect and analyze the internal control values would be substantial and the inferences to be made only tangential to our exploration of SARS-CoV-2 sample stability. We feel that the N2 amplification data supersedes the RP result trends.

 The authors are missing perhaps the earliest work on stability of unsupplemented raw saliva and SARS-CoV-2 detection and are likely more relevant than those currently included in the discussion: doi: 10.3201/eid2704.204199 We have added this to the discussion.

 The citation recommended for line 105 also reports on stability of SARS-CoV-2 RNA when cycled through various temperatures, and demonstrating that cold chain transport is not required. We agree.

 The figures would be more informative, if in addition to the averages depicted, if the results for each pair could also be depicted. This would allow the reader to more robustly analyses how the pairs performed.

 Thank you for the suggestion. We have added a before-after plot to show the original and re-test Ct value for each individual. Panel A includes individuals who had an “undetermined” re-test value that was imputed as 40 for the purposes of analyses while panel B removes these individuals entirely. 

 The authors fail to reflect on some of the large changes in Ct values between some of the pairs. Were re-tests double checked? Could anything different have happened during that time (primers, MMX)? Could any samples have not been tested properly the first time? Are any discrepancies more consistent per month perhaps further indicating a slight difference in that first test month?

 The laboratory team strives to maintain quality in both sample handling and data integrity. A large change in Ct values could be defined as +/- 3 Ct’s. The great majority of our Ct deltas were within this threshold. A Ct value difference of 3 could be caused by variance in pipetting, heterogeneity of saliva, freeze/thaw cycling, or lot-to-lot variation of kit reagents. Retests were double-checked and reviewed before being reported, mastermix and primer/probe sets were from the same manufacturer, and differences from month-to-month were investigated with no root cause found to bias the data.

 Line 2 of the supplementary table shows an initial result of 0 – that doesn’t seem to be accurate. It could be helpful to have table either by month or by initial Ct. We appreciate your diligent checking of our data. The 0 was inaccurate and has been updated to the correct number (27.8). All other entries have also been checked and no other errors were found. We have also updated the data so that the observations are sorted by date.

Reviewer 3 CT - Ct Changed

 Line 112: Yale’s SalivaDirect dual-plexed RT-qPCR protocol: please add the reference or link Added

Reviewer 4 Line 115: -80C -� -80ºC Changed

 The presentation of the results could be improved in several ways, most notably by placing the numerical comparisons in a table, rather than presenting them as text.

 Thank you for the suggestion. We have added several tables summarizing the numerical results. 

 The authors have drawn some strange conclusions based on the Ct values for the assays. IN comparing mean Ct values over time, they have not presented confidence intervals for the means, and have made statements about changes which seem likely to me to be “noise.” Furthermore, they seem to suggest that lower Ct values represent lower reliability, which isn’t true at all. They state that “January, the second oldest samples, displayed low correlation and reliability.” While the figure supports this conclusion, the authors provide no explanation for this odd behavior (error in assay performance? Specimen mishandling?). We have added 95% confidence intervals to the figures, tables, and texts for better transparency. Our statements regarding reliability were not derived from the raw Ct values but from the intraclass correlation (ICC) analyses that were mentioned in the methods and results section. We considered the test and re-test to act as sort of raters. An ICC allows us to gauge the reliability of two raters, in this case, the test and re-testing periods. Because the only difference between the test and re-test periods should be that the sample underwent being frozen and stored, we assume that a low ICC, which corresponds to low reliability, is attributable to being frozen and stored. It is from this analysis that we made statements regarding reliability. 

 The Figure could be significantly improved. There is no justification for the lines connecting the point estimates. Each point estimate should be accompanied by confidence intervals. We have added 95% confidence intervals to the means and have removed the lines connecting the points. 

 The authors have described antigen concentration assessment, and presented some verbiage in the results section, without presenting any sort of formal analysis. I suppose the comments are supported by data in the supplementary Excel spreadsheet, but I would be much happier if there were to be a more formal presentation of what the data say. We have added tables and verbiage to represent the analyses involving antigen concentrations. 

Reviewer 5 The paper is concise and straightforward, the conclusions are valid. The only part that is missing is alignment with MIQE guidelines for quantitative real-time PCR (DOI: 10.1373/clinchem.2008.112797). Understandably, the authors used commercial kits. However, if the information on validation of the kits is available, it is worth including it into the manuscript. Added the following citation to the methods section Vogels et al., Med 2, 263–280 March 12, 2021 ª 2020 Elsevier Inc. https://doi.org/10.1016/j.medj.2020.12.010

 Another comment, the mean values in Figure 1 are given without error bars and n values for the number of samples. From supplementary data it is hard to understand how many samples were available for retesting for each month. The significance of the difference in the January sample should be reported by p-value. We have added confidence intervals to the means displayed in figure 1 and have added the number of samples (N) for each month on the x-axis for clarity. 

We appreciate the utility of hypothesis testing, but the study was designed with the intent to focus on reliability through intraclass correlation analyses. As such, we did not select a sample size appropriate for testing the difference in means and do not feel that assigning a p-value to the difference adds any significant meaning to the report. Keeping in line with the American Statistical Association’s suggestions (cited below), we have added confidence intervals to the calculated means and differences in means which we believe adds more value than p-values. 

Citation: Ronald L. Wasserstein & Nicole A. Lazar (2016) The ASA Statement on p-Values: Context, Process, and Purpose, The American Statistician, 70:2, 129-133, DOI: 10.1080/00031305.2016.1154108

Reviewer 6 Ln132: Why would you create a CT value when no value was obtained. These should just not be run on statistical analysis since giving them a point value. See response below

 When observing your data in figure 1 and in the averages, the freeze thaw seemed to improve detection based on the lower CT. Would this be more pronounced if the negative specimens were removed. It is also important to discuss these missed samples and the CT from the initial test. Were they near the LoD.

 Response to this comment and the one above: When a sample had an “undetermined” Ct test value, it had a Ct above the instrument’s detectable limit which was ~40. Instead of excluding these observations that had a high, albeit unknown, Ct value, we chose to impute these values with the maximum Ct, 40. We recognized the potential bias that can be introduced with this mention which is why we conducted sensitivity analyses excluding these observations, as mentioned in the methods and results. We have altered the methods and results section to make it clearer that all analyses using imputed values were followed by sensitivity analyses where those same observations were removed.

 From the data trend, it almost appears that a freeze thaw improves the sensitivity of the assay, which has been a discussion in the field and possible concern for the FDA in evaluating retrospecitive specimens. It would be beneficial if a small subset of new specimens could be frozen and tested after 24 hours of freeze thaw to determine if The authors concur that this would be an important factor when evaluating retrospective specimens. The observations from this study may contribute to later publications of a robust study designed to look specifically at sample stability after multiple short-term freeze/thaws.

 Ln124 and throughout: I would suggest reviewing the manuscript for conversational and indirect language. As an example, ln 124 states” the original chosen 10 could not be re-tested for some reason”. I would remove “for some reason” and add a sentence of the numbers that were not tested and reasons. This continues into more of the methods, which make it a bit unclear of how the specimens were tested. For example, n the sample selection when 10 were taken from December and 10 from January are these tested at the same time so one batch is X-months frozen and the January batch is X-1 months old or were they all stored for 12 months prior to testing? Language has been changed throughout. In the instance that a sample could not be re-tested, it was because there was not enough volume leftover to conduct a re-test. We have clarified this in the methods. We used a random sampling scheme when there were more than 10 available samples for any given month to choose the original 10 and 4-5 backup samples so that chances of selection bias were minimized. All samples were re-tested on the same day (11/18/21) with the idea that we could test viability UP to 12 months. We have added a sentence explaining this to the Sample Selection section.

 Ln104: What is meant by compliance (i.e. was this approval via the institutions IRB)? This has been reworded, these were residuals from surveillance testing, completely deidentified stored residuals, no IRB required.

 I would consider changing the term re-test to thawed specimens or something similar. When I read re-test I am thinking of a possible repeat for a test that was invalid.

 We have changed “re-test” to “thawed specimens” in figures 1 and 2. 

 Link figures in text when indicated and presenting data. Done

 As there is only 87 data points I think it would be interesting to see the N1 CT values as a dots where the samples are on X axis CT on Y and 2 points for each sample so we can see the spread of CT values for individual samples and not as an average. Thank you for the suggestion. Per yours and other reviewers’ requests, we have added a before-after plot to show the original and re-test Ct value for each individual. Panel A includes individuals who had an “undetermined” re-test value that was imputed as 40 for the purposes of analyses while panel B removes these individuals entirely.

---

## [Decision Letter · Decision Letter 1]

22 Jul 2022

PONE-D-22-12527R1SARS-CoV-2 reliably detected in frozen saliva samples stored up to one yearPLOS ONE

Dear Dr. Frediani,

Thank you for submitting your manuscript to PLOS ONE. After careful consideration, we feel that it has merit but does not fully meet PLOS ONE’s publication criteria as it currently stands. Therefore, we invite you to submit a revised version of the manuscript that addresses the points raised during the review process.

ACADEMIC EDITOR: IN addition to the comments raised by the diligent reviewers, please amend the following:1- Full vendor details should include, company, city (state), and country2- Tables should be in landscape not portrait layout3- Conclusion should be in a separate section4- Study strengths and limitations should be in a separate section, headed as addressed. It should be ahead of the conclusion.  

We look forward to receiving your revised manuscript.

Kind regards,

A. M. Abd El-Aty

Academic Editor

PLOS ONE

Journal Requirements:

Reviewers' comments:

Reviewer's Responses to Questions

**Comments to the Author**

1. If the authors have adequately addressed your comments raised in a previous round of review and you feel that this manuscript is now acceptable for publication, you may indicate that here to bypass the “Comments to the Author” section, enter your conflict of interest statement in the “Confidential to Editor” section, and submit your "Accept" recommendation.

Reviewer #1: All comments have been addressed

Reviewer #2: All comments have been addressed

Reviewer #3: All comments have been addressed

Reviewer #4: (No Response)

Reviewer #6: All comments have been addressed

2. Is the manuscript technically sound, and do the data support the conclusions?

Reviewer #1: Yes

Reviewer #2: Yes

Reviewer #3: No

Reviewer #4: Partly

Reviewer #6: Yes

3. Has the statistical analysis been performed appropriately and rigorously? 

Reviewer #1: Yes

Reviewer #2: Yes

Reviewer #3: Yes

Reviewer #4: Yes

Reviewer #6: Yes

4. Have the authors made all data underlying the findings in their manuscript fully available?

Reviewer #1: Yes

Reviewer #2: Yes

Reviewer #3: Yes

Reviewer #4: Yes

Reviewer #6: Yes

5. Is the manuscript presented in an intelligible fashion and written in standard English?

Reviewer #1: (No Response)

Reviewer #2: Yes

Reviewer #3: Yes

Reviewer #4: Yes

Reviewer #6: Yes

6. Review Comments to the Author

Reviewer #1: The authors made a great job in answering raised criticisms

From my side the paper is suitable for a publication

Reviewer #2: I commend the authors for thoroughly addressing each of the reviewers comments. The paper is much stronger and adds valuable data to the literature.

Reviewer #3: "SARS-CoV-2 reliably detected in frozen saliva samples stored up to one year"

No additional comments for the current manuscript.

Reviewer #4: The authors have adequately responded to all of my comments except one - what went wrong with the January specimens? Understanding this seems very important to supporting the conclusions of the paper. Things look good for 8 of 9 months, but this is not a level that I generally would be willing to accept in the clinical laboratory, nor would I expect a regulatory agency to accept as adequate in any part of the developed world. An explanation, or at least a speculation on what went wrong with these samples seems very important to me.

Reviewer #6: Ln 90 what do you mean by simple random samples? I would suggest removing the word simple.

All other comments have been addresed.

7. PLOS authors have the option to publish the peer review history of their article (what does this mean?). If published, this will include your full peer review and any attached files.

Reviewer #1: **Yes: **Mario Plebani

Reviewer #2: No

Reviewer #3: No

Reviewer #4: No

Reviewer #6: No

---

## [Author Response · Author response to Decision Letter 1]

28 Jul 2022

Please see Response to Reviewers document

---

## [Decision Letter · Decision Letter 2]

1 Aug 2022

SARS-CoV-2 reliably detected in frozen saliva samples stored up to one year

PONE-D-22-12527R2

Dear Dr. Frediani,

We’re pleased to inform you that your manuscript has been judged scientifically suitable for publication and will be formally accepted for publication once it meets all outstanding technical requirements.

Kind regards,

A. M. Abd El-Aty

Academic Editor

PLOS ONE

Additional Editor Comments (optional):

Reviewers' comments:

Reviewer's Responses to Questions

**Comments to the Author**

1. If the authors have adequately addressed your comments raised in a previous round of review and you feel that this manuscript is now acceptable for publication, you may indicate that here to bypass the “Comments to the Author” section, enter your conflict of interest statement in the “Confidential to Editor” section, and submit your "Accept" recommendation.

Reviewer #4: All comments have been addressed

2. Is the manuscript technically sound, and do the data support the conclusions?

Reviewer #4: (No Response)

3. Has the statistical analysis been performed appropriately and rigorously? 

Reviewer #4: (No Response)

4. Have the authors made all data underlying the findings in their manuscript fully available?

Reviewer #4: (No Response)

5. Is the manuscript presented in an intelligible fashion and written in standard English?

Reviewer #4: (No Response)

6. Review Comments to the Author

Reviewer #4: (No Response)

7. PLOS authors have the option to publish the peer review history of their article (what does this mean?). If published, this will include your full peer review and any attached files.

Reviewer #4: No

---

## [Editor Report · Acceptance letter]

2 Aug 2022

PONE-D-22-12527R2 

SARS-CoV-2 reliably detected in frozen saliva samples stored up to one year 

Dear Dr. Frediani:

I'm pleased to inform you that your manuscript has been deemed suitable for publication in PLOS ONE. Congratulations! Your manuscript is now with our production department. 

Kind regards, 

on behalf of

Prof. A. M. Abd El-Aty 

Academic Editor

PLOS ONE